# Textural and Consumer-Aided Characterisation and Acceptability of a Hybrid Meat and Plant-Based Burger Patty

**DOI:** 10.3390/foods12112246

**Published:** 2023-06-01

**Authors:** Bjørn Petrat-Melin, Svend Dam

**Affiliations:** Business Academy Aarhus, School of Applied Sciences, 8260 Viby J, Denmark

**Keywords:** acceptability, meat alternative, preference mapping, check-all-that-apply, consumer survey

## Abstract

The hamburger has been targeted for substitution by numerous plant-based alternatives. However, many consumers find the taste of these alternatives lacking, and thus we proposed a hybrid meat and plant-based burger as a more acceptable alternative for these consumers. The burger was made from 50% meat (beef and pork, 4:1) and 50% plant-based ingredients, including texturised legume protein. Texture and sensory properties were evaluated instrumentally and through a consumer survey (*n* = 381) using the check-all-that-apply (CATA) method. Expressible moisture measurements indicated a significantly juicier eating experience for the hybrid compared to a beef burger (33.5% vs. 22.3%), which was supported by the CATA survey where “juicy” was used more to describe the hybrid than the beef burger (53% vs. 12%). Texture profile analysis showed the hybrid burger was significantly softer (Young’s modulus: 332 ± 34 vs. 679 ± 80 kPa) and less cohesive than a beef burger (Ratio 0.48 ± 0.02 vs. 0.58 ± 0.01). Despite having different textural and CATA profiles, overall liking of the hybrid burger and a beef burger were not significantly different. Penalty analysis indicated that “meat flavour”, “juiciness”, “spiciness” and “saltiness” were the most important attributes for a burger. In conclusion, the hybrid burger had different attributes and was described with different CATA terms than a beef burger but had the same overall acceptability.

## 1. Introduction

World population has now surpassed eight billion, and at the same time global meat consumption has steadily increased to a global average of more than 42 kg per capita per year in 2020 [1]. In addition, even though the consumption of meat per capita is beginning to show signs of plateauing in high income countries, there is no indication of the increase in overall global consumption stagnating during the coming years [2]. It is estimated that between a fourth and a third of global anthropogenic greenhouse gas (GHG) emissions are due to food systems. Furthermore, it is well established that intensive animal farming for meat production is associated with substantial GHG emissions [3,4], with ruminants being the largest contributor, predominantly due to the production of methane from intestinal fermentation of ingested biomass. Therefore, due to climate concerns, policymakers in many parts of the world are pushing for a reduction in consumption of animal derived foods, and beef in particular. The aim is to incite consumers to move towards a more plant-forward diet for a reduction of its carbon footprint. The recently updated Danish dietary guidelines were developed to also take into account the climate impact of food systems, an approach that is seen in several other European countries as well, i.e., sustainability is specifically mentioned in the dietary guidelines of countries including Italy, France, Germany, the Netherlands and the United Kingdom [5].

Reducing meat in the diet without replacing it with another protein-rich food will reduce the consumer’s overall protein intake. Legumes are often highlighted as a suitable replacement for meat because most legume seeds have a comparatively high protein content. They do, however, have a less optimal amino acid profile than animal protein [6] and lack some of the micronutrients that are found in meat, as for example vitamin B_12_ which is found almost exclusively in animal foods [7]. Legumes may also contain so-called antinutritional factors that reduce the bioavailability of other nutrients, such as phytate reducing bioavailability of divalent cations and trypsin inhibitors interfering with protein digestion [8]. Thus, legumes, like all raw materials, have limitations, but in combination with, for example animal sourced ingredients, may fulfil functional and dietary requirements.

A flexitarian diet avoids eating meat with many or most meals. In Denmark in 2022, 12% of consumers identified as flexitarian, whereas 75% identified as meat-eaters [9]. There is, however, 48% who would like to reduce their intake of beef. However, many meat-eating consumers adhering to a typical Western diet are reluctant to forego the meat-eating experience in favour of legumes, as illustrated by the low legume consumption observed in Denmark and other Nordic countries [10]. This has led to the implementation of processing techniques that can alter the structure of legume proteins to mimic the texture of meat. One such method is extrusion cooking, where the globular legume seed storage proteins are subjected to high temperature and shear force leading to denaturation, alignment and aggregation into a fibrous structure akin to the myofibrillar protein structure of muscle meat [11]. The resulting product is known as texturised vegetable protein (TVP) and is currently used in many plant-based meat alternatives. Originally, TVP was developed using soy protein, but nowadays many manufacturers use yellow pea, faba bean and/or other protein sources as starting material [12]. The texture of TVP based products can be comparable to minced meat products and in some cases whole cuts of meat, but in surveys consumers find the taste lacking. For example, 48% of respondents (*n =* 1000) in a Danish survey named taste as the primary reason for not buying plant-based alternatives to animal derived products [9], and 68% of respondents (*n =* 1631) in a survey from the United States said that the taste of animal meat was superior to that of plant-based substitutes [13]. Therefore, plant-based meat alternatives can still be considered a niche product in most Western countries [14].

A possible low-threshold route to convince more consumers to reduce their meat intake in favour of more plant-based products could be through hybrid foods. Hybrid foods combine meat and plant-based ingredients to reduce the meat content, and thereby the climate impact, but also retain much of the sensory characteristics of an all-meat product. Previous attempts to market hybrid meat and plant-based products have not been particularly successful [15]. However, the idea of reducing meat consumption in favour of more plant-based alternatives has recently become more widely recognised [16,17].

A hamburger is a convenience, or fast food, product consisting of a meat burger patty, often made from beef, sandwiched between two halves of a bun. It usually also contains some type of dressing and sometimes lettuce, pickled cucumber, tomato and other ingredients. The global fast food market size in 2020 was USD 862 billion, of which the category “Burger & Sandwich” accounted for 36% [18]. Reducing the total amount of beef consumed through burgers thus presents a potentially large global reduction in intake. Therefore, in the present study, we developed a hybrid burger patty consisting of 50% meat and 50% plant-based ingredients. Physical, textural and sensory quality of the burger was evaluated and compared to an all-meat beef burger as reference, through physical and texture analysis, as well as a consumer survey using the check-all-that-apply (CATA) method. To the best of our knowledge, this is the first report of such a study with Danish consumers.

## 2. Materials and Methods

Burger patties: The TVP (made from two-thirds pea and one-third faba bean protein and extruded milled pea starch) was kindly supplied by Crispy Food A/S (Gørlev, Denmark). Additional ingredients used in the patty are shown in Table 1. Unless otherwise indicated, the ingredients used in the patties were household brands purchased from a local supermarket. To make the patties, TVP was mixed with water and non-meat ingredients (gluten (Nutty Vegan, Holstebro, Denmark), red beet juice (Biotta Juices, Fishers, IN, USA), tomato paste, milled extruded pea starch (Crispy Food, Gørlev, Denmark), dried porcini mushroom, salt, monosodium glutamate (Ajinomoto Foods Europe, Paris, France), yeast flakes (Nutty Vegan, Holstebro, Denmark), garlic powder and black pepper), and mixed thoroughly in a stand mixer until a visible gluten network had formed. Bacon (Danish Crown, Randers, Denmark) was added to the mixture which was then processed in a meat mincer fitted with a 6 mm hole plate. The resulting mince was mixed with minced beef and finely chopped frozen coconut oil-in-water emulsion, which was made as follows. Methylcellulose (Special Ingredients, Chesterfield, United Kingdom) (final concentration in emulsion 0.5% *w*/*v*) was mixed with one part water using an immersion blender, followed by slowly adding three parts melted coconut oil until a mayonnaise-like texture was achieved. The emulsion was then frozen and subsequently was finely chopped before use. The estimated nutritional composition of the hybrid burger patties is shown in Appendix A. Data for estimation of nutritional composition was acquired from ingredient and raw material specifications, as well as the official Danish food composition database [19]. The final mixture was shaped into 100 g patties with a diameter of 8 cm and pan-fried to an internal temperature of 75 °C, as were reference patties of identical size and shape made from minced beef. Salt and ground black pepper were added to the reference patties at similar amounts as the hybrid burger patties immediately before frying. Afterwards, the burger patties were either analysed or frozen for later use in the consumer survey. Freezing of the patties for the survey was necessary for logistical reasons.

Texture profile analysis: Circular samples were cut using a 30 mm diameter steel ring. These were then cut to a height of 19 mm. Texture profile analysis (TPA) was carried out on a TMS Pro texture analyser (Food Technology Corporation, Sterling, VA, USA), using a flat cylindrical probe with a diameter of 50 mm. The TPA settings were as follows: trigger force = 2 N, compression and return speed = 2 mm/s, strain *=* 60% of initial height, with no pause between compressions. The attributes calculated were as follows: Young’s modulus = stress/strain (kPa); cohesiveness = ratio of work performed during second and first compression, respectively; springiness = ratio of sample height at second compression to initial height. Note, Young’s modulus was chosen because it is independent of sample dimensions and therefore is more generalisable [20].

Cooking loss: Hybrid and beef burger patties were pan-fried to 75 °C internal temperature. The cooking loss was calculated as the percentage of weight lost after cooking.

Expressible moisture: Expressible moisture was assessed using the procedure described by Earl et al. [21] with minor modifications. Briefly, approximately 3 g pieces of cooked burger patty were placed in a small cup fashioned from one inner layer of Whatman no. 50, 70 mm and three outer layers of Whatman no. 3, 50 mm filter papers. These were placed inside a 50 mL centrifuge tube and centrifuged at 15,000× *g* and room temperature for 15 min. Expressible moisture was taken as the percentage of weight lost after centrifugation of the cooked patty.

Consumer survey: We conducted a consumer survey at a foodservice expo held over three days in March 2022 in Herning, Denmark. The survey was structured as a CATA questionnaire including a nine-point hedonic liking scale for each product anchored at “dislike very much” and “like very much” and with five meaning “neither like nor dislike”, as described by Ares and Jaeger [22]. The CATA question included 20 descriptive terms generated by a focus group of students from Business Academy Aarhus’ Food Technology and Application programme (*n =* 35; M = 16; F = 19). The terms, which are shown in Table 2, were printed in randomised order on the survey questionnaire to minimise primacy bias [23]. The questionnaire also prompted participants to describe their perceived ideal burger patty using the same 20 CATA terms which permitted identification of important attributes through penalty analysis [22]. In addition to the CATA and liking questions, the participants were asked to state age and sex, and whether or not they were a regular consumer of plant-based meat alternatives. Questionnaires from 381 assessors were filled out correctly with regard to the CATA and liking questions and were therefore included in the analysis. The assessors were recruited among visitors to the expo and were asked to evaluate the hybrid and beef burgers and fill out the questionnaire at a table behind the expo booth under ambient conditions. The samples were served single-blinded labelled with a three-digit code and in randomised order on white paper plates after reheating to 75 °C over a water bath in a standard household oven. Assessors were instructed to taste and check terms for one burger patty at a time and were given tap water for palate rinsing between samples. The terms for the “ideal” burger patty were checked at the end.

Climate impact (CO_2_-eq): The climate impact of the beef and hybrid burgers were estimated in terms of CO_2_-equivalents (CO_2_-eq) per burger in a use case exemplified here by a standard cheeseburger. The CO_2_-eq for the different ingredients were obtained from The Big Climate Database [24], which is the official Danish database for climate impact of food products. In the cases where a specific entry for an ingredient was not available in the database, a similar product was chosen instead, e.g., wheat flour substituted for gluten in the hybrid burger patty, and Danbo cheese 45+ substituted for burger cheese.

Data analysis: The results of the texture profile analysis, cooking loss and expressible moisture were analysed using two-tailed student’s *t*-test. Hedonic liking of the test burgers was analysed using the paired Wilcoxon signed rank test, and differential use of CATA terms for hybrid, beef and ideal burgers was analysed using Cochran’s Q test.

Correspondence analysis was carried out on the contingency table of CATA term use from the questionnaire, and a biplot was created to assess relationships between terms and samples.

Hierarchal clustering was performed using the CLUSCATA algorithm in XLSTAT-Sensory [25]. After clustering, all assessors with incomplete demographic data were excluded. Differential liking of the burgers in the generated clusters was tested using two-way non-parametric ANOVA with interaction on Aligned Rank Transformed data according to Wobbrock et al. [26]. The factors were “cluster” and “type of burger” and assessor was included as random effect. Potential differences in cluster demographic composition were assessed using *Χ*^2^-test on counts.

Penalty analysis was carried on the CATA and liking data from the questionnaire to assess the importance of attributes for consumer acceptability of the burgers.

Differences were considered as significant at the 0.05 level, and all analyses were carried out in R version 4.3.1 [27] and XLSTAT Sensory 2022.2.1 (Addinsoft, Paris, France).

## 3. Results

### 3.1. Texture Attributes

In addition to taste, the texture and mouthfeel of food has a profound impact on consumer acceptability. The texture of soft solid foods can be measured instrumentally using TPA, which involves a two-cycle compression of uniformly cut samples of the food under investigation. A number of attributes can be calculated from the force–time and stress–strain curves, each of which approximate different aspects of the sensory experience of eating the food [28,29]. The results from the TPA are shown in Table 3. The Young’s modulus of the beef burger was two-fold higher than that of the hybrid, indicating that the beef burger would possibly be experienced as somewhat firmer than the hybrid. The beef burger’s cohesiveness was twenty percent higher than the hybrid’s, suggesting that the hybrid burger will require less total energy input during mastication before swallowing. In contrast, the springiness was marginally higher for the hybrid burger; however, a difference of such limited magnitude will in all likelihood not be detectable by the consumer. Taken together, the hybrid burger will presumably be experienced as softer and easier to chew. Cooking loss measures the amount of moisture lost during frying. The hybrid burger lost markedly less moisture than the beef burger (Table 3) and, as expected, the subsequent amount of expressible moisture was correspondingly higher for the hybrid burger. This should translate to a juicier sensory experience when eating the hybrid burger.

### 3.2. Climate Impact

The amounts and estimated CO_2_-eq footprint for each ingredient in the regular cheeseburgers is shown in Table 4. The CO_2_-eq footprint of the hybrid burger patty is less than half that of the beef patty. This is because not only is the amount of beef in the hybrid patty reduced by 50%, a further 8.5% is exchanged with bacon, which has a lower CO_2_-eq footprint than beef (4.8 vs. 33 kg CO_2_-eq/kg). The CO_2_-eq footprints for each cheeseburger sums to 1.80 and 3.60 kg CO_2_-eq for the hybrid and beef burger, respectively. Thus, replacing the beef patty in a cheeseburger with the hybrid patty halves the overall GHG emissions associated with the burger.

### 3.3. Consumer Survey, Evaluation of Hybrid and Beef Burgers

The assessors in the survey were recruited at an expo for foodservice providers in Denmark. After exclusion of incomplete or incorrectly filled out forms, 381 assessors were included in the analysis. Assessor demographics are shown in Figure 1.

#### 3.3.1. Overall Liking

On a discrete nine-point scale anchored at “dislike very much” and “like very much” the consumers’ scores for the two burgers were not significantly different, as assessed by the Wilcoxon paired signed rank test (*p* = 0.32). The respective liking scores (±s.e.) were 5.4 (±0.1) for the beef and 5.3 (±0.1) for the hybrid.

#### 3.3.2. CATA Term Usage

The assessors were asked to check any of the 20 CATA terms they felt described the burger they were tasting at a given moment. The frequency of term usage is shown in Table 5. Cochran’s Q test was employed to determine which terms’ use was significantly different for the meat and hybrid burger, which was most of the terms, except “Metallic”, “Pepper”, “Meat colour”, and “Brown surface”. Thus, there is independence between rows and columns, indicating that the assessors experienced them as having different sensory characteristics. For six of the twenty attributes the difference in usage frequency between the meat and hybrid burger was more than 3.5-fold. These were “Soft”, “Dry”, “Juicy”, “Tough”, “Pink”, and “Off-flavour”. The hybrid burger was perceived as softer, less dry and juicier, more tender, and pinker than the beef burger but was also perceived by 29% of the assessors as having an off-flavour, in contrast with just 3% indicating this for the beef burger.

The relationship between the products being tested and the CATA terms was explored using correspondence analysis. A graphical representation is shown in Figure 2. There is a clear separation of the two tested burgers as well as the ideal burger. Apparently, neither of the tested burgers represented an ideal burger to the assessors, which according to the biplot, is associated with a juicy, fat, spicy and salty taste and has a pink appearance with a crust. The hybrid burger was more associated with being soft and having off-flavour, whereas the beef burger was perceived as more firm, tough, well done and dry.

#### 3.3.3. Penalty Analysis

Penalty analysis can be used to determine how much the overall liking of a product drops when the usage of a specific CATA term is different for a product relative to the “ideal”. This provides a tool for research or for product development whereby it is possible to identify product attributes that are either detrimental to consumer acceptability, if present, or are required for optimal acceptability. An overview of the penalty analysis carried out for the hybrid and beef burgers tested in the present study is shown in Figure 3. The graph shows the mean drop associated with certain terms and how large a proportion of assessors had checked this term differently for product vs. “ideal”. The vertical line shows the cut-off at 20% of assessors, above which a given term is deemed relevant for further consideration. Thus, terms in the upper right corner of the graph are the most important for the product being tested. In this case “Meat flavour”, “Spicy”, “Juicy”, “Salt”, “Crust”, and “Pink” can be considered must-have attributes, and conversely “Rubberlike” and “Dry” can be considered must-not-have attributes. The terms “Soft”, “Dark surface”, “Off-flavour”, “Grainy”, and “Tough” could be considered somewhat important for acceptability but should not be prioritised over those that are must-have and must-not-have. For the rest of the terms in Figure 3 either the proportion of assessors or the penalty is too small for them to be considered relevant.

#### 3.3.4. Cluster Analysis

Using the CLUSCATA hierarchal clustering algorithm, the assessors were assigned to clusters where within-cluster dissimilarity between assessors’ CATA term usage was minimised. This resulted in two clusters and a K+1 cluster containing the assessors that did not fit either of the two obtained clusters. Of the original 381 assessors, after exclusion of assessors with incomplete demographic data, 92 were assigned to cluster 1 and 138 to cluster 2. The relationship between the burgers and the CATA term usage is visualised by correspondence analysis-generated biplots Figure 4B,C for clusters 1 and 2, respectively. Assessors in both clusters apparently associate the beef burger with “Well done”, “Firm”, “Tough”, and “Dry”, and an ideal burger with “Pepper”, “Pink”, “Salt”, “Crust”, and “Spicy”. Conversely, the hybrid burger is more associated with “Rubberlike” and “Grainy” in cluster 1, where in cluster 2 the association with “Soft” and “Juicy” is strongest. Analysis of the liking scores (Figure 4A) for the two clusters showed that assessors in cluster 2 scored both burgers higher, and preferred the hybrid over the beef burger. In contrast, the assessors in cluster 1 preferred the beef over the hybrid burger, although this difference was not significant. There was a non-significant trend (*p* = 0.08) for fewer women than men in cluster 1 and more women than men in cluster 2. The ratio of regular plant-based eaters to non-eaters also tended to be higher in cluster 2 than in cluster 1; however, it was also not significant (*p* = 0.11).

## 4. Discussion

The current and growing demand for meat and other animal foods is unsustainable with regard to climate impact [30], with beef possibly being the largest single contributor, when factoring in the volumes consumed [31]. This is also reflected in the EAT-Lancet commission recommendations of consuming just 14 g/day of beef, lamb or pork [32]. Hamburgers are a popular beef-based food and are therefore an excellent target product category for introducing a reduction in beef consumption. Nonetheless, the beef in a burger patty has distinct technological and sensory characteristics that should be mimicked, such as texture, mouthfeel and meaty and fatty flavours. Replacing the burger patty with one that is plant-based and more climate friendly is not sensorially acceptable to many consumers [10,11]. The use of legume protein can be associated with “grassy”, “beany”, bitter” and “astringent” off-flavours from volatile compounds [33]. For example, a CATA-based study by Neville et al. [34] compared hybrid burgers and sausages to fully plant-based and fully meat-based products and reported that “processed appearance” and “off-flavour” together with lack of “meat flavour” were important attributes for acceptability of burgers. In addition, some consumers may have an affective connection with meat, and therefore the idea of giving up meat in favour of a plant-based alternative is associated with negative emotions [35]. Thus, they continue eating all-meat burgers. However, a discrete choice analysis survey with German and Belgian consumers indicated that meat-hybrid foods would be preferred over fully plant-based foods. The authors concluded that meat-hybrid foods could help facilitate the transition to a more plant forward diet [17].

The hybrid burger patty developed here had a nutritional composition comparable to a beef burger, albeit with a slightly lower fat but a higher carbohydrate content and some dietary fibre at 1 g per 100 g of burger patty. A study evaluated the effects of substituting TVP made from pea, sunflower, or pumpkin protein for 30% of the meat in pork meatballs on nutritional and climate impact characteristics [36]. It was found that fibre and unsaturated fatty acid content was increased. The former was similar to the present study, and the latter was due to the use of canola oil. We used coconut oil as the source of plant-based fat. Coconut oil contains almost exclusively saturated fatty acids, and therefore the unsaturated fatty acid content in our hybrid burger is likely to be low, although it was not specifically analysed. Compared to preparation of a beef burger patty, preparation of the hybrid burger patty requires several additional steps, as described in Section 2, which will make scale-up challenging. In the present lab-scale form of the procedure the coconut oil emulsion is prepared separately and requires freezing, and ingredients are mixed and minced separately. This makes scale-up unlikely, and thus the procedure for this proof-of-concept hybrid burger patty should be simplified to be industrially relevant.

The hybrid burger was evaluated both instrumentally and in a consumer survey. In order to imitate the texture of beef, TVP was combined with gluten as a binder and secondary textural element and with extruded pea starch as a water and fat binder. Similar texture properties to a beef patty were not achieved, as revealed by the TPA. However, thirty percent of assessors in the survey prefer a soft burger, as indicated by the frequency of “Soft” checked for the ideal burger; however, this will need to be within an acceptable range, and not too soft, as 21% checked “Firm” as a preferred attribute for an ideal burger. The penalty analysis also indicated that softness is a desired attribute. The moisture-related attributes of the burgers showed that the hybrid burger would provide a much juicier mouthfeel than the beef burger, which was also supported by more than half of the assessors checking “Juicy” for the hybrid burger. Juiciness is the most desired attribute, with an 82% checked frequency for an ideal burger. The improved juiciness of the hybrid burger is a result of the lower observed cooking loss, which translated to more moisture released during mastication, in line with the higher expressible moisture measurement. The underlying molecular mechanism is likely the lower content of connective tissue in the form of collagen in the hybrid burger, and possibly the water-holding properties of the added extruded pea starch. Loss of moisture during cooking is largely a result of collagen contraction due to heat [37], and since the hybrid burger contains less meat and therefore less collagen, less moisture is expelled.

The CATA questionnaire indicated that the burgers were perceived as dissimilar, presumably owing to the textural and moisture-related differences discussed above. That taste and flavour related terms differed was perhaps not entirely surprising, as it is well-known that legume proteins and volatile compounds can result in various off-taste and off-flavours [38]. The CATA demonstrated that the hybrid burger was not an indistinguishable replacement for a beef burger; however, it could be an acceptable one, as judged by the similar overall liking score. In the study by Neville et al. [34] the liking scores for two different hybrid burgers were statistically the same as for a beef burger, as in the present study. Conversely, in their correspondence analysis the hybrid burgers were clustered together with the meat burger, and thus were associated with the same CATA terms, which was quite different in our analysis. However, this was likely caused by the inclusion of two fully vegetarian burger variants in their study, and the dissimilarity between their hybrid and vegetarian burgers was more pronounced than between their hybrid and meat burgers. In our correspondence analysis the different use of the CATA terms is clearly illustrated. To assess this further we performed a penalty analysis to learn which attributes should be prioritised when developing a burger patty. The penalty analysis corroborated the previous results that pointed to “Spicy”, “Juicy”, and “Salt” as important attributes, but also “Meat flavour”, which incurred the largest penalty when absent. Based on this, and since the hybrid burger was already favourably evaluated in juiciness, effort should be put into enhancing spiciness, saltiness, and meat flavour to accommodate consumer preferences.

An interesting question to ask could be whether it is a specific type of consumer that embraces a product like the hybrid burger, and if so, what separates them from other consumers. In an attempt at answering this question we conducted a cluster analysis using the CLUSCATA hierarchal algorithm resulting in two separate clusters of assessors. The clustering is carried out solely on the use of the CATA terms in the survey, and therefore also provides some insight to the question of whether there is a connection between term usage and liking. Cluster 2 assessors liked both burgers more than cluster 1 assessors and preferred the hybrid over the beef burger. Cluster 2, in contrast to cluster 1, trended towards being composed of more women than men. Historically, eating meat has been associated more with masculinity than femininity [39], which may help to explain this observation. Furthermore, Bush and Clayton recently reported that there is a significant gender difference in attitudes toward climate change [40], which is more pronounced in high income countries. They showed a clear correlation between a country’s GDP and the gap between women’s and men’s perceived seriousness of climate change, with women being more concerned than men. Even though the burgers were presented in a blinded manner, only labelled with three-digit codes, it is likely that some assessors were able to guess which was the hybrid and which was the beef burger, which may then have influenced their attitude towards them. The correspondence analysis of the term usage shows a distinctly different pattern in the two clusters. In cluster 1 the hybrid burger is more associated with terms that have a negative impact on liking, whereas in cluster 2 the association with more positive or neutral terms was stronger. Along with the penalty analysis, this serves to underpin an apparent connection between term usage and overall liking of the burgers.

Potential weaknesses of the study are mainly associated with the consumer survey. Firstly, for logistical reasons it was not possible to prepare the burger patties on site, which meant they had to be cooked in advance, then frozen and subsequently thawed and reheated at the expo where the survey was carried out. Nevertheless, this method is also used in fast food restaurants in Denmark. In addition, in order to comply with food safety regulations, they were cooked to an internal temperature of no less than 75 °C, which may be higher than what some consumers would prefer regarding colouring, texture and mouthfeel. However, the two burgers received identical treatments, so whatever negative effect the above may have had would be the same for both.

## 5. Conclusions

The trend towards a more plant-forward diet as a tool for climate change mitigation has shown signs of stagnation. This may reflect lacking sensory acceptability towards plant-based meat alternatives and an emotional attachment to meat among the consumers who have not embraced the trend. Therefore, we hypothesised that a hybrid burger might be a less challenging change of habitual diet for this consumer segment. We developed a hybrid burger patty that was softer and juicier but associated with an off taste as well as less meat flavour than a beef burger. However, consumers found the hybrid burger overall as acceptable as a reference beef burger. The climate footprint, expressed in CO_2_-eq, is also substantially reduced in the hybrid burger. Our observations point to specific critical sensory attributes to develop for possibly increasing acceptability of a hybrid plant/meat burger patty. Cluster analysis only indicated trends toward defining distinct consumer characteristics that could predict liking and CATA term usage for the hybrid burger. In conclusion, we describe here proof-of-concept that a 50%/50% plant/meat hybrid burger may be an acceptable substitution for a meat-only burger, however, this should be evaluated more directly through willingness to purchase and behavioural studies.

## Figures and Tables

**Figure 1 foods-12-02246-f001:**
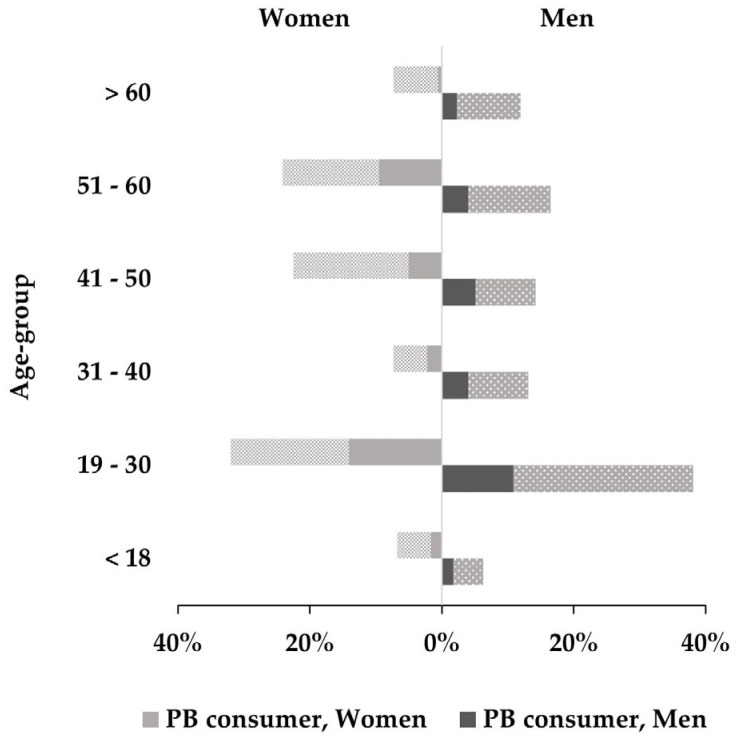
Age and gender distribution of assessors in the consumer survey. Shaded part of each bar shows proportion of assessors who sometimes eat plant-based meat alternatives.

**Figure 2 foods-12-02246-f002:**
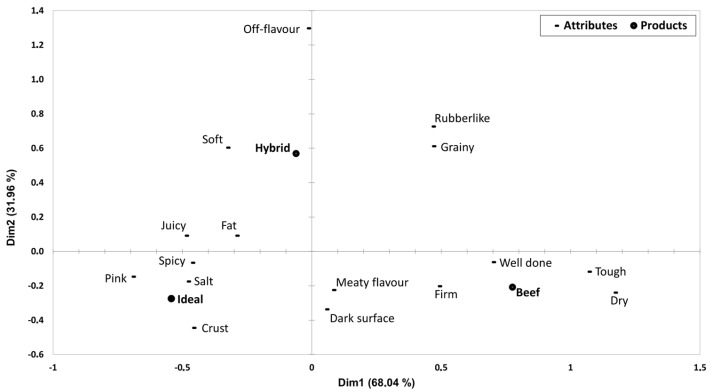
Biplot of rows (terms) and columns (burgers, including ideal) from correspondence analysis of the CATA data. Only terms with significantly different usage are shown (*p* ≤ 0.05).

**Figure 3 foods-12-02246-f003:**
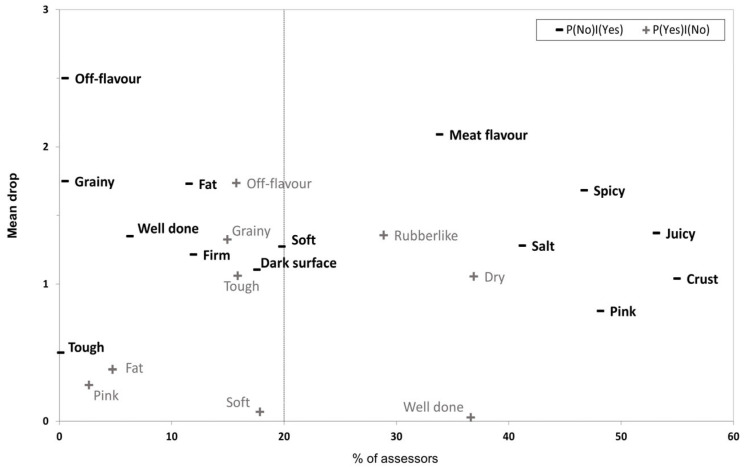
Mean drop in overall liking score for a burger when a CATA term was either checked for ideal and not for tested burger (−P(No)|(Yes)), or not checked for ideal and checked for tested burger (+P(Yes)|(No)).

**Figure 4 foods-12-02246-f004:**
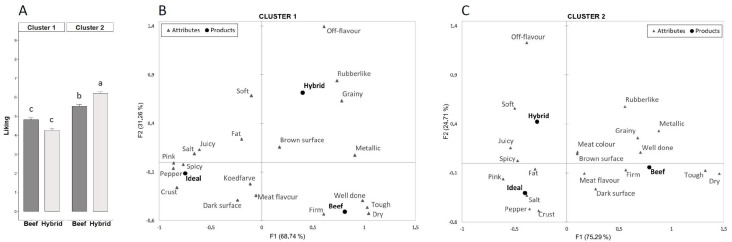
Mean liking scores for the two burgers in two clusters (panel **A**) generated by the CLUSCATA method. Different letters above bars indicate statistically significant difference (*p* < 0.05). (panels **B** and **C**) Biplot of rows (terms) and columns (burgers, including ideal) from correspondence analysis of cluster 1 and 2, respectively.

**Table 1 foods-12-02246-t001:** Ingredients used in hybrid (50%/50% plant/meat) and beef burger patty.

Ingredient	Percentage	
	Hybrid	Beef
Minced beef (14–18% fat)	41.5	-
Minced beef (12% fat)	-	98.5
Water	28.0	-
Smoked salted bacon	8.5	-
Gluten (from wheat)	6.2	-
TVP ^(1)^	4.0	-
Red beet juice	2.5	-
Tomato paste	1.2	-
Milled extruded pea starch	1.2	-
Dried porcini mushroom	0.3	-
Salt	0.6	1.3
Monosodium glutamate	0.4	-
Yeast flakes	0.3	-
Garlic powder	0.1	-
Ground black pepper	0.1	0.2
Coconut oil	1.5	-
Methylcellulose (E461)	<0.1	-

^(1)^ Texturised pea and faba bean protein in 2:1 ratio.

**Table 2 foods-12-02246-t002:** Consumer-generated sensory terms for check-all-that-apply survey of hybrid vs. beef burger patty.

Texture	Taste	Appearance
Soft	Fat	Pink
Dry	Salt	Well done
Rubberlike	Meat flavour	Meat colour
Tough	Metallic	Dark surface
Grainy	Spicy	Brown surface
Juicy	Off-flavour	
Firm	Pepper	
Crust		

**Table 3 foods-12-02246-t003:** Physical attributes related to texture and sensory perception. Results are given as means ± s.d. (*n =* 6).

		Sample		
Attribute	Unit	Hybrid	Beef	Significance ^(a)^
Young’s modulus	kPa	332 ± 34	679 ± 80	***
Cohesiveness	Ratio	00.48 ± 0.02	00.58 ± 0.01	***
Springiness	Ratio	00.77 ± 0.01	00.76 ± 0.01	*
Cooking loss	Percent	17.2 ± 0.2	26.9 ± 0.1	***
Expressible moisture	Percent	33.5 ± 0.1	022.3 ± 0.03	***

^(a)^ Asterisks indicate statistically significant difference between samples: * = *p* < 0.05, *** = *p* < 0.001.

**Table 4 foods-12-02246-t004:** Climate impact of ingredients used in a regular cheeseburger.

Ingredient	Amount (g)	CO_2_-eq per Burger (kg)
Bun	50	0.04
Patty	100	1.39 ^a^/3.19 ^b^
Salad	13	0.01
Tomato	27	0.02
Pickled cucumber	17	0.03
Cheese	27	0.21
Mayonnaise	17	0.02
Dressing	17	0.08

^a^ Hybrid; ^b^ Beef.

**Table 5 foods-12-02246-t005:** Frequency (%) of assessors’ (*n =* 381) use of CATA terms to describe burgers and their ideal burger.

Attribute	Hybrid	Beef	Ideal	Significance ^(a)^
Soft	51	6	30	***
Dry	11	62	0	***
Rubberlike	36	22	0	***
Tough	7	25	0	***
Grainy	18	13	1	*
Juicy	53	12	82	***
Firm	15	44	21	***
Crust	11	17	67	**
Fat	11	5	15	**
Salt	20	11	53	***
Meat flavour	33	64	72	***
Metallic	4	6	0	n.s.
Spicy	30	12	62	***
Off-flavour	29	3	1	***
Pepper	9	9	46	n.s.
Pink	21	3	57	***
Well done	26	63	14	***
Meat colour	25	29	34	n.s.
Brown surface	35	34	38	n.s.
Dark surface	7	17	21	***

^(a)^ Cochran’s Q test for significance of difference between use of terms for each sample. Asterisks indicate statistically significant difference between samples: * = *p* < 0.05, ** = *p* < 0.01, ********* = *p* < 0.001, n.s. = *p* > 0.05.

## Data Availability

The data presented in this study are available on request from the corresponding author. The data are not publicly available due to the participants of this study not giving written consent for their data at an individual level to be shared publicly.

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
