# Peer review of "Textural and Consumer-Aided Characterisation and Acceptability of a Hybrid Meat and Plant-Based Burger Patty"

_foods, 2023, doi:10.3390/foods12112246_

Round 1
Reviewer 1 Report
The paper is focused mainly on the sensorial evaluation of a hybrid burger where meat has been partially substituted for legumes protein. I think the authors did a good job in the sensorial analyses and the discussion section, but the physicochemical data require a more profound revision. In fact, my main concern is that I miss a better description of the used ingredients (percentage, commercial brand, oleogel preparation) in order to make the experiment reproducible. I include some comments to improve the quality of the paper:
Line 78 The information about the experiment should be complete in order to allow reproducibility. In that sense, commercial brands of ingredients of burgers should be included, especially TVP, Extruded milled pea starch, red beet juice, mushroom, tomato paste, etc.
Line 83: how a gluten network was formed if no wheat was present?
Line 85: Describe better the formation of the oleogel with coconut oil
Line 87: the composition is not well referenced.
Line 89: reference patties were only minced beef? Which composition? Salt and ground pepper were added at the same concentrations as the reformulated patties?
Where is called “Table 1. Nutritional content….” in the text? The table should appear after it has been called in the text. Are the values of table 1 the means? please include the standard deviation. Were there differences between the two formulations significant?
Line 99: Why hardness parameter was not considered in the texture analysis? Maybe it is one of the most important in meat products. Please consider including the reference of the textural analysis followed.
Line 129: There are two tables 1?
Texture attributes results do not have line numbers. The first sentence is in cursive, why? Which burgers were firmer? I am quite surprised that the deviations were so minimal in texture parameters like cohesiveness and springiness but not in Young´s module.
Line 162: the information from Table 3 should indicate the reference database used to calculate the CO2 footprint.
Line 184: So, There were no differences between the two formulations? Both were equally preferred?
Line 224_ if rubberlike, and dry are considered must-not-have attributes, why in figure 2 are marked with a + not a negative? I am not familiarized with penalty analysis, could explain this analysis better, considering that the readers are also not familiar with it.
Line 238 and line 243: reference…
Line 273: no statistical analysis was applied to the nutritional content of both formulations to consider whether there was an increase in dietary fiber content in hybrid burgers.
I like the discussion section, it has been well aborded.
The English style and grammar is adequate.
Reviewer 2 Report
This research deals with a study that evaluates the quality and characteristics of a burger consisting of 50% meat and 50% plant-based ingredients and its comparison to an all-meat beef burger as a reference through physical and texture analysis, as well as a consumer survey using the check-all-that-apply (CATA) method.
The topic is within the scope of the Journal and very interesting, the manuscript is well-written, and the analyzed parameters are interesting. However, minor revisions are requested. Please see the following comments.
Materials and Methods
Please indicate the methods you used to determine parameters from Table 1.
Lines 87. Please replace "Error! Refer-ence source not found" with "Table 1".
The table "Consumer-generated sensory terms for check-all-that-apply survey of hybrid vs. beef burger patty" should be numbered as Table 2.
Line 118. Please replace "Table 1" with "Table 2".
Table 2 should be renumbered as Table 3. Please add a space before the table caption.
Please replace "Table 2" with "Table 3" in both places mentioned in subsection 3.1.
What do you mean by "but will feel similar with regard to size and shape during mastication". Please formulate for a better understanding.
Please replace "Table 3" with "Table 4" in the table caption and Line 159.
Lines 166 and 167 cover the text.
On a scale from 1 to 9, 5 means "neither like nor dislike".
Please replace "Table 4" with "Table 5" in the table caption and Line 192.
At the bottom of Table 5 please add n.s. = p ≥ 0.05.
Lines 205-206. Please replace "Error! Ref-erence source not found" with "Figure 2".
Figures are not numbered correctly or in the right place in the manuscript. Please revise them.
Figure 2 should be moved above subsection 3.3.3.
Please replace "Figure 3" with "Figure 2" in Lines 218 and 228.
Line 253. Please replace "Figure 4" with "Figure 3" in the table caption.
Line 238. Please replace "Error! Reference source not found" with Figure 3.
Lines 243-244. Please replace "Error! Reference source not found" with Figure 3.
Section 5 "Conclusions" should appear before Line 347.
Reviewer 3 Report
I am very grateful you for the invitation to review manuscript foods-2403021 by Petrat-Melin and coauthors "Textural and consumer-aided characterisation and acceptability of a hybrid meat and plant-based burger patty”. The aim of this study was to develop a hybrid burger patty consisting of 50% meat and 50% plant-based ingredients. The quality and characteristics of the burger was evaluated and compared to an all-meat beef burger as a reference, through physical and texture analysis, as well as a consumer survey using the check-all-that-apply (CATA) method. The work is interesting but needs adjustments to increase the quality of the material.
Comments:
- Line 7: Specify better the “highest climate footprints”.
- Line 10: The sentence “as a lower threshold alternative for these consumers” is not clear.
- The issue of the “climate footprint” must be reviewed, because despite claiming this all the time, they continue to use animal meat. Authors should highlight incentive issues to reduce the consumption of this type of food.
- Abstract: If the “climate footprint” is so important, the authors must present the results of production and carbon neutralization, for example, for products.
- Lines 14-15: What was the determined moisture? This must be presented here.
- Lines 15-16: The results of this parameter must be presented for comparison.
- Lines 19-20: There are no results that determine this. Comparison with a standard product and indicating that they are willing to modify dietary habits for replacement are entirely different aspects.
- Abstract: The conclusion should indicate only points related to the objectives of the study, not assumptions.
- Line 21: Change the repeated keywords by different words from the title.
- Lines 25-28: Indicate per capita consumption.
- Introduction: Authors must include information on the definition of hamburgers and highlight the consumption that can justify the use of this food matrix.
- Lines 42-45: It is also necessary to highlight the limitations of the use of vegetables in general, which is often the limiting factor (no raw material is 100% advantageous). A critical discussion of this needs to be included.
- Lines 45-47: This is not clear. Some countries have low meat consumption for cultural reasons. Please specify this point better.
- Introduction: The question of flexivegetarianism, applied to this context, is not pointed out.
- Line 67: What is the sentence “have seen mixed results”?
- Line 71: Specify the parameters referring to the sentence “quality and characteristics of the burger”.
- 2. Materials and Methods, Burger patties: Please include the ingredients in a Table to facilitate the understanding of what was included in each composition.
- Lines 87-88: Check the sentence “in Error! Reference source not found.”
- In the sentence: “indicating that the percieved firmness of the burgers would possibly be experienced as somewhat different from each other”: Is the greater firmness for beef hamburger?
- Line 156, 3.1. Texture attributes: Include numerical results for comparison throughout the text.
- Line 118: This is Table 2. Review the sequence of Tables throughout the text.
- 3.1. Texture atributes: Table 2 is Table 3.
- Lines 158-164: This information should be detailed more clearly. The smaller carbon footprint for bacon should be more detailed.
- Lines 205-206: Check the sentence “in Error! Reference source not found.”
- Line 228: All figure numbering is wrong or missing.
- Lines 238; 243 Check the sentence “in Error! Reference source not found.”
- Lines 258-261: The information has already been presented previously. Use the discussion to add new and important information to the work.
- Lines 262-264: The authors must indicate the important technological aspects that are part of this food matrix, as the substitution of ingredients will impact such properties.
- Line 266: Indicate which negative sensory properties are reported.
- Lines 270-273: The information has already been presented previously. Use the discussion to add new and important information to the work.
- Lines 274-276: Is the verified concentration sufficient for this? It is important to discuss the intake requirements and the amount to achieve the physiological effect.
- Lines 277-281: The information has already been presented previously. Use the discussion to add new and important information to the work.
- The discussion is not used for comparison with other products similar to what was produced here.
- The discussion is not used to indicate the technological changes caused and are important for this type of product.
- The discussion of sensory attributes is also repeated. Please include information about what is expected, interaction, and known attributes provided by plant proteins, among others.
- Authors must include a conclusion item highlighting the points observed in relation to the objectives of the work.
Round 2
Reviewer 1 Report
I appreciate the answers to my comments. I think the paper is more focused on the sensory tests and the physicochemical information provided is not correct since no analyses have been performed to verify the composition. In fact, from the information in Table 1, many of the texture parameters can be related to the inclusion of more water in the hybrid sample and this will affect the sensory parameters.
Table 1 should appear in the material and methods section after it has been cited in the text. I recommend deeply revising the guidelines of the journal.
Line 131: Texturized pea and faba protein in a 2:1 ratio means the ratio between pea and faba or water to hydrate. The formulation is not well described. Also, the oleogel should be an ingredient as a whole (made with water methylcellulose and coconut oil). Where is the water used in the oleogel. Which amount of water was used to hydrate pea and faba flour? There are still many inconsistencies. For example, were red beet juice and tomato paste commercial? Which composition? Brands should be included. The experiment is still not reproducible. The water content has to be very different from the hybrid to the control sample and of course, this fact will also impact the caloric content. At least the authors should have measured moisture content in samples.
Line 149: If no physicochemical analysis was done, table 2 should be eliminated. I am not sure how can guess the fat content of control or beef patties. For example, if the meat had 14-18 % fat (table 1) how in Table 2 consider only 12 % . The fat percentage is very variable according to the meat cut used for the patty elaboration. Considering the authors used coconut oil, saturated fat seems too little. To claim the improvement of the nutritional content of hybrid samples, the composition should be verified by physicochemical analyses. Otherwise, this information should be eliminated and also the related discussion.
Line 154: although the authors have included similar concentrations, they should describe both formulations (hybrid and control).
OK
Reviewer 3 Report
The authors have improved the manuscript satisfactory.
Author Response
We sincerely thank reviewer number 3 for their efforts.